# Heparin Resistance in Patients Receiving Extracorporeal Membrane Oxygenation: A Review

**DOI:** 10.3390/jcm13247633

**Published:** 2024-12-14

**Authors:** Tatyana Li, Azhar Zhailauova, Aidyn Kuanyshbek, Iwan Wachruschew, Shaimurat Tulegenov, Vitaliy Sazonov, Timur Kapyshev

**Affiliations:** 1Department of Anaesthesia and Intensive Care, Heart Center CF “University Medical Center”, Astana 010000, Kazakhstan; aidynks1@gmail.com (A.K.); ivanvahrushev08@gmail.com (I.W.); shaimurat_2011@mail.ru (S.T.); 2Department of Surgery, Nazarbayev University School of Medicine, Astana 010000, Kazakhstan; vitaliy.sazonov@nu.edu.kz (V.S.); timur.kapyshev@nu.edu.kz (T.K.)

**Keywords:** extracorporeal membrane oxygenation (ECMO), heparin resistance, coagulopathy, complication

## Abstract

Heparin resistance (HR) in patients on extracorporeal membrane oxygenation (ECMO) exacerbates bleeding and thrombogenesis. Thus far, there is no universal definition of what this condition entails and no unified strategy for assessing heparin’s efficacy in ECMO patients. The most frequent discrepancy when it comes to defining HR is the difference in the reported doses: units per day (U/d) or per kilogram per hour (U/kg/h). Another disagreement arises with regard to the various methods of measuring unfractionated heparin (UFH) efficacy. Due to numerous processes that begin with ECMO initiation, including protein layer formation on the surface of circuits, the recruitment of immune cells, the activation of complement and contact activation systems, and platelets, assessing pure antithrombin consumption is complicated. Moreover, there is an alternative anticoagulation procedure performed by a serine protease inhibitor named heparin cofactor II, which could also contribute to heparin consumption. Considering simultaneously launched processes of inflammation and thrombogenesis in response to contact with artificial surfaces on ECMO, we listed the possible mechanisms contributing to additional antithrombin consumption. The effect of the flow on the platelets’ activation and von Willebrand factor (vWF) assembly was also described. We reviewed the scientific literature from PubMed and Embase to identify possible definitions of heparin resistance during ECMO treatment among pediatric and adult cohorts. We identified 13 records describing different approaches to assessing HR and described our vision of delineating HR on ECMO.

## 1. Introduction

Extracorporeal membrane oxygenation (ECMO) is a continuous non-physiological blood transportation system that consists of artificially coated tubes and an oxygenator that has an extended area of contact with the blood [1,2]. In addition, the pressure generated by a centrifugal pump changes the physiological pattern of the blood flow [3]. When blood comes into contact with foreign surfaces, numerous effects are observed, such as protein adsorption and the activation and precipitation of the complement system, thrombocytes, and leukocytes along the extracorporeal circuits, leading to inflammation and thrombogenesis [4,5,6,7,8]. Bedside anticoagulation, which is usually achieved through the titration of unfractionated heparin (UFH), is essential, as it interferes with thrombogenesis along the circuit, in the oxygenator, and in the pump itself [9]. However, extended ECMO exploitation requires the long-term use of heparin, which may ultimately lead to heparin resistance (HR) due to the depletion of antithrombin as a result of the inflammatory response and coagulopathy [10]. The higher heparin doses required to meet the necessary anticoagulation parameters may cause undesirable bleeding [10]. Hence, to achieve a satisfactory balance between bleeding and thrombosis on ECMO, a deep understanding of the changes in hemostasis is required. In this review, we aim to comprehensively describe the condition of HR in patients on ECMO from the molecular and clinical perspectives. Furthermore, we summarize the currently available data on the incidence of HR among pediatric and adult cohorts on ECMO. Also, we suggest modification and extension of the current description of HR in patients on ECMO based on the collected information.

## 2. Issues Associated with Defining Heparin Resistance

Heparin resistance is a condition in which standard doses of UFH are unable to facilitate satisfactory levels of anticoagulation. However, a more precise definition has yet to be established [11]. The initial ambiguity in defining heparin resistance in patients treated with ECMO arises from the definition of standard UFH doses, as dosages can be measured in units per day (U/d) or in units per kilogram per hour (U/kg/h), with the latter taking the patient’s weight into account. According to a recent survey conducted by Levy et al., the most popular UFH dose considered indicative of HR is >35,000 U/d [12], though other researchers have identified the presence of HR at UFH doses >40–70 U/kg/h [13,14]. These discrepancies introduce significant uncertainty with regard to the definition of a standard dose of UFH. Another ambiguity arises due to the different goals of heparin administration. Cardiac surgery patients on cardiopulmonary bypass (CPB) and intensive care patients require different loading doses; thus, defining a standard UFH dose leading to HR becomes more complex. Patients on CPB demand an activated clotting time (ACT) > 480 s. If this target cannot be achieved at a UFH dose of 500 U/kg, HR is suspected [15]. In comparison, in patients on ECMO, the anticoagulation target is an ACT span within 180–220 s after initial UFH rates at 7.5–20 units/kg/h, followed by further elevation to achieve a therapeutic effect at the rate of 20–50 units/kg/h [16]. For comparison, if an adult weighing sixty kilograms receives a dose of 50 units/kg/h per day, the total daily dose of heparin reaches 72,000 U/d, which is two-times higher than the current definition of HR involving a dose of 35,000 U/d.

Further uncertainty emerges from the coagulation measurement methods. ACT cannot be used to measure clot strength but is applied as an instantaneous assessment of fibrin clot formation at a certain moment in time. The concentration of intrinsic pathway components, the availability of platelets’ phospholipids to activate coagulation, the temperature, and the ACT machinery all influence ACT, leading to deviations when it comes to assessing the efficacy of heparin. However, ACT is appropriate for assessing the response to heparin in dynamics [17]. Despite the quick response offered by the ACT method, traditionally, the activated partial thromboplastin time (aPTT) and anti-Xa assay have been the preferred methods of assessing heparin’s efficacy, and a recent survey showed that the aPTT and chromogenic anti-Xa assay had almost equal applicability for UFH monitoring [12]. Although the anti-Xa assay is generally preferred, aPTT is still widely applied in critically ill patients on ECMO. The chromogenic anti-Xa assay measures the direct inhibition of Xa by heparin, reflecting its sole anticoagulation effect. During the test, heparin binds to the patient’s antithrombin (AT) and inactivates a fixed amount of an external Xa. The inactivated Xa then couples with the chromogen in the luminescent reaction, allowing for the direct deduction of the amount of Xa that is inactivated by the heparin–AT complex. However, the anti-Xa assay does not assess the formation of the clot itself. In patients receiving ECMO, when UFH is administered, anti-Xa activity should be maintained at 0.3–0.7 IU/mL [18]. If the anti-Xa level is less than 0.3 IU/mL, it is necessary to administer supplementary heparin with a 10–20% dose elevation [17]. If the anti-Xa level remains subtherapeutic, the patient should be checked for signs of AT replenishment or transitioned to another anticoagulant such as direct thrombin inhibitor (DTI) [18]. aPTT, in turn, assesses the time it takes to form a clot after the activation of factor XII until the formation of the clot [17,19]. This test assesses the functionality of the intrinsic and common pathways, as well as the effect of heparin on the coagulation cascades. Factor VIII and fibrinogen are overexpressed during the inflammatory state; thus, the true anticoagulant activity of heparin could be masked in critically ill patients under the condition of pseudo heparin resistance, where, due to elevated factor VIII and fibrinogen, aPTT values demonstrate subtherapeutic concentrations [19]. In patients on ECMO, aPTT values should be in the range of 60–90s [17]. If a subtherapeutic aPTT level persists, the anti-Xa level should be checked, and appropriate management should be undertaken [18].

## 3. Incidence of Heparin Resistance

We searched the PubMed and EMBASE databases for the following medical subject headings (MeSH): “heparin resistance”/exp OR “heparin resistance” AND “extracorporeal oxygenation”/exp OR “extracorporeal oxygenation”. In response to our request, we received a total of 65 studies. Before screening, we removed three duplicates and nine conference abstracts. We screened the remaining 53 records. Of these, 26 records were deleted due to several factors: the definition of HR in non-ECMO patients, the definition of bivalirudin resistance, the application of anticoagulants other than UFH, and systematic reviews. The other 27 records were screened, and subsequently, 9 papers were excluded due to the absence of proper HR criteria. Thus, 18 studies were included. We extracted 18 records, including original studies, case reports, and case series, which described the phenomenon of heparin resistance in patients on ECMO who were initially prescribed UFH, which was then replaced or used in combination with another anticoagulant approach if necessary. Studies with incomplete data were excluded. A flow chart summarizes the selection strategy (Figure 1). Selected studies are illustrated in the table below (Table 1).

The majority of studies determine their own thresholds for the identification of HR in patients on ECMO, which vary substantially. The threshold for the UFH dose is based on the respective subtherapeutic anti-Xa or aPTT values. We divided our findings into two parts that consisted of eight pediatric and nine adult studies. The first study mentions the development of HR in a 12-year-old child suffering from cardiogenic shock. HR was manifested as an inability to reach the therapeutic anti-Xa concentration at a UFH dose of 25 units/kg/h and as the formation of thrombi in the circuit. To ensure that proper anticoagulation was achieved, the patient was switched to bivalirudin [20,23]. The next case report illustrates the implantation of ECMO in a 2-h-old newborn neonate due to respiratory and cardiac deterioration [21]^.^ At a UFH dose of 55 µg/kg/h, the corresponding anti-Xa level was < 0.1 U/mL. The AT activity was 31%; after the elevation of AT activity up to 50%, therapeutic anti-Xa values were not achieved and the patient was switched to bivalirudin, with successful attainment of the anticoagulation goal. A potential reason for the developed HR is the immaturity of AT molecules in newborns, which prompts the application of alternative anticoagulation under ECMO. The largest pediatric study mentioned in our review defines HR under ECMO as UFH > 40 IU/kg/h or when a therapeutic anti-Xa level is unattainable [14]. The study analyzed the remaining deficiency in AT after its replenishment. Of 2028 AT measurements from 191 pediatric patients (median age of 65 days), 43.3% showed signs of AT deficiency whilst they were on ECMO [14]. After AT replenishment, 36% of cases remained AT-deficient, which emphasized the importance of establishing an AT replenishment protocol if UFH is selected as the anticoagulant agent. AT replenishment was performed in neonates at AT < 50%, while in the elder cohort, it was administered at AT < 80%. The next study compared UFH and bivalirudin anticoagulants administered during ECMO treatment; the heparin group showed a higher incidence of bleeding (observed in 44% of cases vs. 12.5%) but fewer events of thrombosis (observed in 11% of cases vs. 25%) [22]. In comparison to the UFH group, the bivalirudin group showed longer maintenance of the desired aPTT (44% in UFH vs. 65% in bivalirudin group). It should be noted that the median age of the UFH group was 4 months, while that of the bivalirudin group was 0.4 months. Neonates have decreased AT levels, and it may take up to six months for the AT structure to fully develop [37]. Thus, some hospitals prefer to use bivalirudin as a safe alternative to UFH in infants. In this study, HR was defined as a failure to reach the required anti-Xa level when the UFH dose was 26 IU/kg/h [22]. Out of 27 pediatric patients on UFH, only 1 developed HR and was subsequently switched to bivalirudin. Another pediatric study described case series with the development of HR in a 12-year-old patient placed on ECMO due to COVID-19-initiated inflammatory syndrome. The study does not define a specific UFH dose for HR but states that the anticoagulant values were subtherapeutic regardless of AT and fresh frozen plasma supplementation [23]. The second pediatric study that compared bivalirudin and UFH anticoagulation under ECMO initiated AT replenishment when anti-Xa and aPTT were subtherapeutic and AT activity was less than 60% [24]. The comparison showed that the bivalirudin group attained the required anticoagulation parameters faster with less bleeding and thrombogenicity. As previously stated, the hospital prefers to use bivalirudin due to the inadequately mature AT molecules in young patients and bivalirudin’s lower cost [24]. The seventh pediatric study emphasized that it may be necessary to increase the dosage of recombinant AT supplementation to reach an AT activity of 80–120% in their pediatric cohort with a median age of one month [25]. HR was defined in the study as no increase in ACT at a UFH dose > 40 IU/kg/h. The last pediatric study reports HR in a 3-month-old patient at a UFH dose reaching 50 IU/kg/h with a corresponding ACT value of 120 s [26]. The patient was successfully switched to argatroban, and therapeutic anticoagulation was attained. The largest study on HR in adults defined HR in terms of the dose per day and the dose per patient’s weight, e.g., >35,000 IU/d and >20 IU/kg/h, respectively. The incidence of HR reached 23% (47/197). Notably, in this study, HR was not associated with thrombogenesis; rather, increased thrombogenicity was associated with the VA-ECMO type and with COVID-19 [27]. Another study that defined HR described a case in which UFH at the maximal dose of 43,200 IU/d was observed in a patient diagnosed with COVID-19 and placed on ECMO due to respiratory failure [28]. The required aPTT and ACT were reached only after argatroban was prescribed as an additional anticoagulant therapy. In a study by Raghunathan et al., half of the 67-patient cohort experienced HR for at least one day during ECMO therapy [29]. HR was defined as a UFH ≥ 35,000 IU/d. Interestingly, there was no difference in bleeding, thrombosis, or survival between those with and without HR [29]. An increase in the infusion rate of heparin was successfully implemented to manage all cases of HR [29]. Another study that supported a patient-related agreement between aPTT and anti-Xa when the fibrinogen level and factor VIII were greatly elevated (7.5 g/L and 606%, respectively) defined HR in the adult cohort as UFH > 35,000 IU/d, with an HR incidence of 33% [30]. The next case report demonstrated the efficacy of alternative treatment with bivalirudin when there was a failure to reach therapeutic anticoagulation (anti-Xa: 0.38 IU/mL) using UFH at a dose of 32 IU/kg/h with 90% AT activity [31]. One more presentation of HR in a patient with ARDS and with subtherapeutic anti-Xa at a UFH dose > 50,000 IU/d was successfully resolved upon transition to bivalirudin anticoagulation [32]. In an investigation that compared HR in pediatric and adult cohorts simultaneously, HR was defined as a subtherapeutic anti-Xa level under UFH doses of 23.6 and 15.3 units/kg/h, respectively. AT replenishment improved the anti-Xa concentration, but no significant reduction under UFH infusion was observed [33]. The study that compared the management of HR through either escalating the UFH or through the replenishment of fresh frozen plasma defined HR as UFH > 24,000 IU/d [34]. Interestingly, the mean daily dose of heparin for the UFH group was almost two-times higher than that for the fresh frozen plasma group (19405.6 vs. 10669.1 IU/d) [34]. The two remaining studies defined HR as UFH > 60,000 and >38,000 IU/d in cases of pneumosepsis and ARDS that were successfully treated with conversion to anticoagulation using direct thrombin inhibitors [35].

In the case of a pediatric population, bivalirudin is preferred in some hospitals as a non-inferior and safe alternative to UFH due to the physiologically reduced availability of mature AT molecules in these younger patients. Meanwhile, for the adult population, bivalirudin is applied in the case of failure to reach therapeutic anticoagulation parameters if HR or heparin-induced thrombocytopenia occur. Moreover, a study that examined AT activity after replenishment stated that, among 191 pediatric patients, almost one-third remained AT deficient [14]. A study considering rAT replenishment in children suggests that more frequent examination of AT and doses of AT replenishment higher than those proposed by the manufacturer are beneficial [25]. In contrast, among the adult population, two studies demonstrated the resolution of HR upon increasing the UFH infusion rate [29,34].

## 4. Origins of Heparin Resistance

Although the nature of HR is multifaceted, in most cases, it is defined by a failure to achieve the anticipated effect of UFH in patients as a result of underlying thrombogenesis, inflammation, and the increased depletion of AT [12]. AT participates in many biological processes that may lead to its reduction; for example, it is the main heparin-coupling molecule that achieves a widespread anticoagulant effect. Heparin acts by unfolding the structure of AT, a serine protease inhibitor. This conformational change significantly increases the inactivation effect of the heparin–AT complex on factor Xa, thrombin, and several coagulation factors, thereby impeding clot formation [38]. Mainly targeting Xa and thrombin, to a lesser degree, the heparin–AT complex inhibits factor IXa, XIIa, and VIIa in vitro and in ex vivo mouse models [39,40]. When UFH is administered, the physiological expenditure of AT takes place. Only approximately one-third of UFH mixed chains contain the pentasaccharide domain that can bind AT, reinforcing its anticoagulant effect [41,42]. Early hemostasis experiments showed a thirty percent decline in AT concentration in 24 patients undergoing heparin treatment, regardless of the primary level of AT [43]. Initial AT concentrations were restored only several days after the completion of the treatment. In addition to its anti-thrombogenic activity, AT has anti-inflammatory effects and interacts with both heparin sulfate and endogenous glycosaminoglycans [44]. Prostaglandins produced by the endothelium upon contact with AT decrease the extent of leukocyte extravasation and attenuate thrombocyte activation [45,46,47,48]. AT interacts with the heparan sulfate chain of transmembrane syndecan-4 proteoglycan located on the surface of neutrophils and slows down their chemokine-induced migration [49]. Furthermore, AT’s involvement in mitigating NF-κB transcription leads to the reduced expression of inflammatory mediators such as IL-6, TNF-alpha, and tissue factor (tF) schl [44]. Thus far, we have listed the factors that contribute to AT consumption; however, there are also several mechanisms that are responsible for heparin availability. The physiological neutralization of UFH by plasma proteins was observed in the experiments with low- and high-affinity heparin towards factor Xa [50,51]. UFH molecules with molecular weights higher than 6000 Da can bind plasma proteins. When low-affinity heparin molecules were added, high-affinity heparin molecules were displaced, achieving the required anticoagulation in a dose-dependent manner that stabilized after six hours [50,51]. These observations show that the reversible binding of UFH represents an essential anticoagulation quotient when it comes to stabilizing the response towards continuous heparin infusion in terms of the diverse extent of binding towards plasma proteins. Heparin availability also depends on the state of the thrombocytes’ activation. When a platelet is activated, the released substance platelet-factor 4 counteracts the heparin molecule [52]. Under the flow generated by ECMO, platelets are activated, the longer the duration, the higher the extent of platelet activation [4,53]. Thus, more platelet-factor 4 molecules interact with heparin, contributing to UFH consumption. In a series of cases, patients undergoing CPB with a platelet concentration exceeding 600 mcL exhibited HR [52]. Several studies have also proposed a high preoperative platelet level as a potential prognostic parameter for patients developing postoperative HR [3]. In addition to their well-established anticoagulant activity, endogenous heparin possesses anti-inflammatory properties. Endogenously, heparin is produced and released by basophils and mast cells [54]. Experiments conducted with glycosaminoglycans released from rat peritoneal mast cells demonstrated reduced concentrations of leukocytes in the peritoneum, as well as the decreased migration of leukocytes towards inflammatory stimuli [55]. Tang et al. used a mouse model to demonstrate that heparin hinders the HMGB1-mediated entrance of bacterial lipopolysaccharides (LPS) into the cell, preventing the activation of caspase 11, which leads to the lysis of the cell [56]. A similar outcome was proposed in a human model [57]. One of the largest ACUTE II studies assessed the anti-inflammatory property of UFH and demonstrated a decrease in c-reactive protein (CRP) after the initiation of UFH treatment [58]. Hence, heparin expenditure also occurs for anti-inflammatory purposes. Finally, there is an equally important HR concept that depends on false diagnosis. As mentioned earlier in this review, acute phase proteins expressed during the inflammatory response, such as fibrinogen and factor VIII, may reduce aPTT measurement while a therapeutic dose of heparin is attained [52]. A study involving 13 patients undergoing pulmonary endarterectomy titrated the UFH concentration, relying on postoperative aPTT and anti-Xa assay values. One-third of aPTT measurements showed a decline in laboratory values, suggesting HR, in comparison with the anti-Xa assay, although therapeutic dosages of UFH were achieved [59]. Therefore, in the presence of concomitant elevated heparin consumption due to thrombogenesis and inflammation in patients on ECMO, an anti-Xa assay remains the preferable choice for measuring the effect of heparin, since it does not rely on the concentration of acute phase proteins [19]. False laboratory ACT and aPTT values may also be influenced by hypothermia, hemodilution, a low platelet count, and aprotinin administration [10,60].

### 4.1. Heparin, Antithrombin, and Heparin Cofactor II

The heparin molecule is a negatively charged glycosaminoglycan with an average molecular weight of 15 kDa. UFH is a mixture of glycosaminoglycans in which the molecular weight varies from 5 kDa to 30 kDa, i.e., 5–35 units, obtained from animals’ intestinal mucosa, which contains plenty of mast cells [10]. The basic function of heparin is to enhance the anticoagulant effect of AT. Only one-third of UFH mixtures possess a pentasaccharide string to bind AT. The minimal sequence length of the heparin molecule to bind AT, Xa, and IIa is 18 subunits. A sequence with a length < 18 units can bind Xa but not thrombin [10]. UFH interacts with Xa and IIa through AT in a one-to-one ratio [58]. The diversification of heparin molecules in a UFH mixture, regardless of the normalization from the manufacturer, could be a potential cause of varying patient responses [10]. Most of the anticoagulant activity of heparin is antithrombin-dependent. However, antithrombin-independent anticoagulant activity also exists and is achieved through heparin cofactor II (HCII) [10]. HCII is a serine protease inhibitor that binds thrombin in a one-to-one ratio and accelerates its activity 10 times [61,62]. Three-quarters of the whole anticoagulant activity is attributed to AT, while the rest is conducted through an AT-independent pathway, i.e., via HCII facilitation [63]. Although most of the literature considers the condition of heparin resistance from the perspective of antithrombin consumption, further investigation is needed to identify the unique role of HCII when prolonged exposure to heparin is required.

### 4.2. Material of ECMO Linings

ECMO linings consist of a plastic named polymerized vinyl chloride (PVC), which is a synthetic chemical that is fused with an ester plasticizer such as diethylhexyl phthalate (DEHP) to ensure the flexibility of circuits [64,65]. In this mixture, DEHP comprises up to 40% by weight [65] and is released into the blood circulation [66]. DEHP may lead to carcinogenic, reproductive, and developmental toxicity, and its use should be carefully considered in terms of the potential exposure of neonates or pregnant and breastfeeding patients [66,67]. The search for safer alternatives is currently underway and requires further scaling [65].

In most ECMO tubes, the inner surface of the tube linings is covered with heparin to reduce inflammation and thrombogenesis [3]. Several studies have described the effect of heparinized ECMO circuits. A recent case report demonstrated the development of heparin-induced thrombocytopenia (HIT) in a patient with acute respiratory distress syndrome (ARDS). HIT occurred on the twelfth day after ECMO initiation, leading to thrombocytopenia and the formation of HIT antibodies, although UFH was not introduced into circulation [68]. Although the thrombocytes normalized after the ECMO circuit were replaced, the patient did not survive. A heparinized coating for ECMO has proven beneficial for patients predisposed to bleeding. A very recent transplantation case series demonstrated the heparin-free placement of a patient on ECMO as a bridge to lung transplantation [69]. These cases demonstrated the successful management of ECMO performance without UFH infusion, although the circuits of ECMO were heparinized and AT activity was accurately monitored to avoid HR [69].

### 4.3. Plasma Components and Oxygenator

A membrane oxygenator constitutes the most extensive area of contact with the blood, which may reach 0.8–2.5 m^2^ [70,71]. Currently, oxygenators are made either from polymethylpentene or polypropylene material, while the prior generations were silicon-based [72]. Polymethylpentene is preferred, since polypropylene fibers result in plasma leakage when a gas exchange takes place in the membrane oxygenator [7,73]. Most of the studies that investigated the interaction between the blood and the oxygenator considered an oxygenator covered with polymethylpentene material [74]. Early studies showed that a silicon membrane oxygenator precipitates fibrinogen, albumin, IgG, platelets, fibronectin, and von Willebrand factor (vWF) [16,75]. Later studies involving polymethylpentene oxygenators were shown to precipitate fibrin, platelets, erythrocytes, leukocytes, and vWF on the surface of the oxygenator [5,6,7]. Polymethylpentene is a hydrophobic material [76]; although this enhances the gas exchange in membranous oxygenators, the membrane’s hydrophobic material serves as an adsorption platform for nonspecific proteins that undergo further conformations, making the protein layer more prone to thrombosis [77].

## 5. Coagulation on ECMO

### 5.1. ECMO and Fibrinogen Adsorption

Early interactions between blood proteins and biosurfaces started with experiments between plasma proteins and glass [78]. When plasma proteins come into contact with glass, immediate fibrinogen adsorption on the surface of the glass favors further the precipitation of clotting factor XII, high-density lipoprotein, kininogen, albumin, IgG, and complement system components [78,79]. ECMO tubes are made of PVC mixed with an additive—DEHP. One of the first studies to describe the interactions between blood proteins and PVC states that after twenty minutes, 30% more fibrinogen precipitates on the surface of the PVC–DEHP material in comparison with a well made solely from PVC [8]. Notably, the association between the percentage of DEHP and the amount of adsorbed fibrinogen within the PVC–DEHP surface was linear [8]. On the contrary, albumin adsorption was higher for the pure PVC material compared to the PVC–DEHP mixture. Animal experiments were also conducted; a study on an ECMO sheep model demonstrated a decline in the fibrinogen level, which was successfully restored over a two-day period [80]. These experiments demonstrate that contact between blood and the ECMO lining may induce a decline in circulating fibrinogen. The next issue is to assess the association between ECMO-induced hypofibrinogenemia and bleeding. CPB tubes consist of the same PVC material mixed with DEHP [81]. In a study including 50 children, the mean fibrinogen concentration in the mediastinal bleeding group after CPB was lower compared to the non-bleeding control [82]. A recent retrospective study analyzed adult and pediatric data from 24 patients on ECMO [83]. Almost half of the patients demonstrated a reduced fibrinogen activity of <100 mg/d in some period during ECMO placement [83]. Bleeding events and thrombogenesis occurred in patients who were connected to ECMO for a more extended period, especially for periods lasting longer than 250 h [83]. The studies show evidence of bleeding in patients with decreased fibrinogen levels and function on ECMO that may be related to adsorption to the circuits. Furthermore, the trend of the higher rates of bleeding and thrombogenesis was also associated with a longer period of time spent on ECMO [83]. With the formation of the protein stratum, the further adhesion and activation of platelets and leukocytes along with the components of the complement system affected the existing protein layer, leading to the release of inflammatory cytokines, including IL-6, IL-8, and TNF-alpha, and furthering the inflammatory state [84]. A hydrophilic high-molecular-weight surface coating also prevents nonspecific protein binding due to its branched structure and hydrophilic nature [85]. The currently preferred heparin coating successfully decreased thrombin expression via direct antithrombin binding, reducing the inflammatory state in the circulatory system [86].

### 5.2. Coagulation Factors

The plasma contact activation system, which is composed of factor XII, kininogen, and prekallikrein, plays a critical procoagulation role when blood constituents interact with foreign surfaces. The negative charge situated on the surface of the ECMO coating activates factor XII [77], which then transforms prekallikrein into kallikrein [87]. In turn, kallikrein amplifies the formation of activated FXII, reinforcing the intrinsic coagulation cascade. Subsequently, circulating high-molecular-weight kininogen is cleaved by kallikrein, and the bradykinin molecule is released. Bradykinin, in return, contributes to the permeability of vessels and their vasodilation, enhancing inflammation [87]. Simultaneously, factor XIIa activates factor XI, which commences the sequential activation of the intrinsic pathway.

Renné et al. conducted a series of experiments in mice that showed that the absence of factor XII does not lead to bleeding, but rather weakens the density and quantity of thrombi upon stimulation with collagen and adenosine diphosphate (ADP) [88]. Experiments conducted by Cai et al. used extracorporeal shunts and assessed clot formation in rats subjected to dose-dependent siRNA against factor XII. The results showed that as the siRNA concentration increased, the weight of the clots in the extracorporeal shunt decreased. This signifies the role of factor XII in the initiation of clot formation upon exposure to extracorporeal surfaces [89]. Larsson et al. demonstrated that the application of antibodies against factor XIIa reduced thrombogenesis when rabbits were connected to ECMO [90]. In comparison to heparin, anti-XIIa antibodies provided a similar thromboprotective effect but with less bleeding than was observed after heparin addition [90]. Thus, anti-XIIa antibodies seem to be a potential solution for achieving a thromboprotective effect without bleeding.

### 5.3. Tissue Factor

Tissue factor (tF) is a key initiator of the extrinsic coagulation cascade. Under artificial circumstances, six hours of exposure to an ECMO flow resulted in a manifold increase in tF expression on the surface of monocytes [91]. Recent findings suggest that a change in flow influences tF expression in leukocytes [4]. In vitro ECMO involving heparinized blood demonstrated statistically significant increases in the amount of tF on the surface of the leukocytes under low (0.3 L/min) and high flow (0.7 L/min) [4]. Fischer et al. demonstrated that the incubation of heparinized human blood with foreign biomaterials induced surface tF expression on leukocytes in a time-dependent manner [92]. Furthermore, the extent of tF expression was also linked to the activation of the complement system and other granulocytes. Interestingly, in a study with VA-ECMO settings, the tF pathway inhibitor activity was two-times higher on the first and second cycle after ECMO initiation, unlike in the control group [93]. The tF pathway inhibitor is a physiological anticoagulant that is released by the vascular endothelium, which exhibits anticoagulant activity towards Xa and, in combination with Xa, inhibits tF/VIIa activation [94]. Experiments show that ECMO initiation intensifies tF expression on leukocytes and launches an anticoagulant system to prevent tF-induced extrinsic pathway inhibition. On the contrary, the heparin coating of ECMO linings demonstrated reduced inflammation by decreasing the procoagulant activity of monocytes towards a decline in tF expression by seventy percent compared to the non-heparin coating of ECMO linings [95]. This supports the role of the heparin precoating of ECMO linings in reducing thrombogenesis and inflammation.

### 5.4. Von Willebrand Factor

On ECMO, another significant component of hemostasis that becomes dysregulated is vWF. vWF is assembled in endothelial cells and megakaryocytes [96], and it forms multimers composed of up to 60 molecules [97]. Upon further fusion of vesicles carrying multimeric vWF molecules, vWF molecules organize themselves into longer bundles that bind platelets and collagen [97]. Under increased shear stress that exerts a centrifugal roller pump, vWF multimers are mechanically disassembled and certain vWF domains are exposed to cleavage by ADAMTS-13 metalloprotease [98]. Smaller vWF multimers fail to bind and dock platelets through the exposed collagen to the site of injury, interfering with platelets’ contribution to clot formation and leading to bleeding. Ineffective vWF functioning, which results in bleeding, is defined as acquired von Willebrand syndrome (AVWS) [99]. An assessment of high-molecular-weight vWF multimers in 18 patients during ECMO demonstrated a decreased quantity of high-molecular-weight vWF subunits, as well as a reduced ability to bind collagen on the first day of ECMO [100]. Seventeen patients developed bleeding that was reversed after ECMO removal and after appropriate treatment with cryoprecipitate or thrombocytes [100]. A recent study of a COVID-19 patient connected to ECMO showed a similar decrease in the number of high-molecular-weight vWF multimers that were restored after ECMO removal and treatment with cryoprecipitate, while the heparin concentration remained unchanged [101]. In one study, to correct bleeding caused during AVWS without aggravating hypercoagulation after cryoprecipitate was administered, the patient’s bleeding ceased after the addition of recombinant vWF concentrate [102]. According to the findings of a recent systematic review, all ECMO runs are accompanied by AVWS that develops almost immediately after the ECMO launch and discontinues after explantation [103]. These data suggest that the cause of bleeding should be reconsidered, and the management of this symptom could be extended by considering cryoprecipitate or recombinant vWF concentrate, before changing the heparin therapy.

### 5.5. Platelets

Platelets are an essential part of thrombogenesis. They contain surface glycoproteins, for which the GPIb-IX receptor is of primary importance, facilitating platelets’ attachment to an injury site and subsequent thrombus formation. This GPIb-IX receptor comprises GPIbα, GPIbβ, and GPIX components, with GPIbα being the most superficial and binding to numerous ligands, including VWF, HMWK, FXI, FXII, P-selectin, integrins, and thrombin [104]. When the endothelium is damaged, collagen and vWF are exposed on the surface [105]. Liberated vWF binds to collagen and GPIbα, applying a pulling force that induces conformational changes in glycoproteins, and the subsequent mechanical signal transduction activates platelets [104]. The greater the pulling force applied by vWF, the greater the amount of platelet activation that occurs, according to the expression of activation markers such as P-selectin and phosphatidylserine on platelet surfaces [106]. According to a recent systematic review, almost all ECMO patients experience acquired von Willebrand syndrome. Furthermore, in a study including 107 patients on ECMO, it was shown that the greater the flow power applied, the higher the extent of the shedding of GPIbα glycoproteins from platelets, as evidenced by an increased amount of circulating plasma GPIbα [107]. Also, there was a decline in the concentration of thrombocytes after one hour of ECMO initiation. The study also supported the hypothesis that patients on ECMO show elevated shedding of the receptors and faster removal of platelets from circulation, suggesting accelerated clearance of platelets without the GPIbα surface glycoprotein. A greater extent of shedding also indicates a decline in binding to vWF and reduced docking of the platelets to injured sites, resulting in less effective platelet adhesion and aggregation and, therefore, extended bleeding. Under the constant centrifugal force generated by ECMO pumps, high-molecular-weight vWF multimers disassemble, losing their ability to bind collagen and recruit platelets to the site of bleeding. This phenomenon, as mentioned, is referred to as acquired von Willebrand syndrome [100,103]. The fragmentation of the vWF multimers under shear stress also exposes vWF units to greater degradation by ADAMTS-13 metalloprotease, decreasing the effectiveness of platelet recruitment and activation towards the site of clot formation [108]. Some studies demonstrate that, under flow conditions, more phosphatidylserine molecules (PSs), a marker of activated platelets, were expressed on the surface of platelets in comparison to non-dynamic thrombocytes [4]. A study including the blood of ten patients subjected to ECMO supported the idea that the number of activated platelets depended on the duration of the flow [53]. Platelets subjected to flow were also more prone to adhering to fibrinogen layers rather than to vWF multimers [53].

## 6. Inflammation Under ECMO

### 6.1. Complement System

Alternative pathway complement systems are launched when C3b and IgGs come into contact with adsorbed proteins on the surface of extracorporeal linings [77]. Considering that non-endothelialized foreign surfaces do not possess inhibitory stimuli for complement inactivation, this process is exacerbated, triggering innate immunity and overall inflammation [84]. ARDS patients on ECMO demonstrated the activation of the soluble membrane attack complexes with the enhanced production of Bb components one hour after ECMO implantation [109]. This suggests that the alternative pathway components play significant roles in the initiation of inflammatory responses in patients on ECMO. Another study that included six neonate patients showed maximal activation of the complement system within the first two hours after the start of ECMO and consistency in complement activation for three days [110].

CPB circuits are made of the same material as the circuits for ECMO [111]. Complement system activation was also observed among nineteen cardiac surgery patients who underwent CPB within hours after the treatment began due to their blood coming into contact with foreign surfaces [112]. Similarly, the activation of the complement system was observed in seventeen patients undergoing dialysis within the first 30 min, with a subsequent increase in acute inflammation markers by the end of the process [113]. Circuits for dialysis are made of PVC and a DEHP additive [114]. It was observed that the degraded complement product iC3b binds to dialysis circuits and favors the adhesion of neutrophils by triggering complement receptor type 3, along with formed anaphylatoxins, thereby amplifying the overall inflammatory response [115]. As seen in these experiments, when plasma comes into contact with artificial surfaces, it activates the complement system via an alternative pathway, increasing the risk of an inflammatory response.

### 6.2. Neutrophils, Monocytes, and Cytokines on ECMO

Neutrophils are among the key components of the interplay of coagulation and the inflammation axis. Receptors of anaphylatoxins such as C3a and C5a are expressed on the surface of neutrophils [116]. The priming of anaphylatoxin C5a with its receptor on the surface of neutrophils was shown to induce neutrophil activation and the expression of tFs on their surfaces [117,118]. Early experiments investigating the influence of ECMO on leukocytes showed that neutrophils and monocytes are activated after incubation with the plasma received from patients on ECMO [119]. The overexpression of CD18 integrin, a molecule that participates in cellular adhesion, on monocytes and neutrophils was already observed two to four hours after the beginning of ECMO activation, and the peak of neutrophil-derived elastase was reached within eight hours [119]. Flow cytometry analysis of the plasma received from nine patients on VA-ECMO revealed an augmented number of immature neutrophils with attenuated production of C5a receptors [120]. Interestingly, there was a rise in the number of myeloid-derived suppressor cells [120]. Examining ELISA data, it was found that there was a concurrent increase in the level of pro-inflammatory and immunosuppressive cytokines such as IL-6, IL-8, TNF-α, and IL-10 [120]. These experiments suggest that, along with the activation of the inflammatory response towards artificial surfaces, the increased activity of the innate immune response could also be a concern, and this might lead to immunity exhaustion. The platelet-rich plasma of COVID-19 patients induced a higher mRNA load of tF inside neutrophils and enhanced the formation of NETs and release of tF inside the NETs’ chromatin structures, further propagating thrombogenesis [121]. Although a study conducted on a sheep animal model revealed the formation of NETosis six hours after launching ECMO, the formation of NETs was not observed in patients sixteen hours and two days after ECMO initiation [106,122,123]. Overall, the data suggest that either the inflammatory stimuli were not sufficient to induce NETosis in the human model, or there is an efficient counteracting immunosuppressive process that works in tandem with the anti-inflammatory properties of heparin and antithrombin.

### 6.3. Summary of Coagulation and Inflammation

We examined the changes in hemostasis and innate immunity and found that the following changes contribute to coagulopathy and inflammation after ECMO initiation.Upon contact with the negative charge of extracorporeal circuits, the activation system triggers an intrinsic coagulation cascade, contributing to thrombogenesis.Contact of blood with ECMO linings and the oxygenator induces fibrinogen adsorption and further protein layer formation.Activation of the complement system propagates inflammation.Activation of leukocytes results in increased expression of tF on the surface of neutrophils and monocytes, release of pro-inflammatory cytokines, and overall inflammation.Platelets become activated, and thus are prone to adhere to the fibrinogen protein layer. Shedding of the glycoprotein receptor on the surface of platelets due to mechanical stress leads to the ineffective binding to vWF multimers and enhanced clearance. In addition, the contribution to thrombogenesis leads to thrombocytopenia.Shear stress and fragmentation of vWF multimers by ADAMTS-13 metalloprotease and the inadequate recruitment of platelets to the site of injury lead to the development of acquired von Willebrand disease, which predisposes patients to bleeding.

## 7. Management of Heparin Resistance

### 7.1. Current Strategy

There is no generalized protocol for assessing the efficacy of UFH in patients on ECMO [17]. Thus, there is no universally defined definition for HR. Currently, the assessment of hemostasis on ECMO does not involve comprehensively measuring the procoagulant activity of the contact activation system, the effect of the adsorbed protein layer, the activation of the complement and immune systems, or the effect of the flow after ECMO initiation. Nevertheless, we suggest that the UFH efficiency in patients on ECMO is patient-dependent. In pediatric ECMO, if UFH is used for anticoagulation, more thorough monitoring of AT is suggested in younger patients [14,31,37]. Several hospitals prefer bivalirudin anticoagulation in patients younger than 3–6 months due to AT immaturity, which is not a concern in older patients [22,24,27]. The replenishment of AT is still subject to debate; however, recent studies have shown that AT replenishment allows the necessary therapeutic anti-Xa to be achieved and facilitates a decrease in the UFH requirement [33,34]. A study that compared 20 patients in whom HR was managed by fresh frozen plasma delivery and by an increase in the UFH dose showed that the mean daily requirement of UFH was significantly lower in the group receiving fresh frozen plasma, constituting 10,669.1 IU/d and 19,405.6 IU/d, respectively [34]. Another study evaluated the replenishment of AT when the AT activity was <50% in 14 adult patients with subtherapeutic anti-Xa concentrations at UFH doses of 15–20 IU/kg/h [124]. Thirteen out of fourteen patients achieved therapeutic anti-Xa levels, although the bleeding rate and platelet transfusion were higher after AT supplementation [124]. Investigations among 191 pediatric patients with a mean age of 65 days revealed that, in 43% of the AT measurement cases, the AT level was lower than the defined threshold for neonates by being < 50%, whereas for elder children, it was lower by being <80% [14]. After AT replenishment (200 doses), 36% of patients (63) remained AT deficient [14]. In patients with suspected HR, 16.6% achieved a decrease in their UFH dose after AT supplementation, while 83.3% required the same UFH dose or higher [14]. This study shows that the pediatric cohort, especially at the median age of 2 months, may need more thorough consideration with regard to AT replacement. One more conclusion that may be drawn from this study is that AT infusion does not necessarily decrease the UFH requirement and thus does not reflect actual heparin responsiveness. Another very recent investigation evaluated the effect of AT consumption in fifty adult patients during 7 days on VA-ECMO [125]. The participants developed an acquired AT deficiency (50% < AT < 70%), which, in most cases, resolved within 3 days. Although there was evidence of AT deficiency, the median AT level increased from 48% to 63% by the end of the 7-day study period. Therapeutic anti-Xa levels (0.3–0.5 IU/mL) were reached upon escalating the UFH dose, and the study observed no correlation between moderate AT deficiency and heparin responsiveness [125]. Nevertheless, the importance of AT supplementation in the perioperative period is well illustrated in a case series of three patients in whom ECMO placement with only heparinized circuits and no actual anticoagulation therapy was a successful bridge to lung transplantation, with a mean AT activity before and after transplantation of 73.3% (91%, 56%, and 73%) and 74.3% (82%, 79%, and 62%), respectively [69]. All the abovementioned examples emphasize the importance of a patient-dependent strategy when it comes to AT supplementation in patients on ECMO.

Considering the treatment of HR, Levy et al. created a highly comprehensive protocol for managing HR in the ICU that emphasizes the need to switch to anti-Xa activity if there is no change in the aPTT on UFH dose elevation [18]. If anti-Xa shows subtherapeutic concentrations even after the UFH dose has increased, and there is evidence of thrombogenesis, the patient’s AT level should be checked and replenished if necessary, or they should immediately be switched to a DTI such as bivalirudin or argatroban [18]. The Extracorporeal Life Support Organization guidelines describe a therapeutic UFH range of 20–50 units/kg/h. The UFH dose used to define HR in the literature varies from 24,000 to 61,000 IU/d. As documented in most of the studies mentioned in Table 1, the maximal UFH dose that still resulted in subtherapeutic anti-Xa or aPTT was defined as an HR dose. To further eliminate the ambiguities associated with defining HR in patients on ECMO, we suggest performing calculations of the UFH dose in terms of the patient’s weight rather than in terms of a daily dose. It would also be beneficial to document the concurrent AT activity at the highest UFH dose with anti-Xa < 0.3 IU/mL to draw any potential associations between clinically useful AT–heparin coupling. Notably, 25% of endogenous anticoagulation is performed by HCII [63]. This may explain how therapeutic anti-Xa can be achieved even when the AT level is low. At high UFH concentrations, HCII can bind the clot-bound form of thrombin [126]. On the contrary, AT can bind only free thrombin molecules [126]. The monitoring of HCII during long-lasting UFH exposure would also be beneficial in quantifying the effect of HR in a unified formula. The effects of artificial surfaces inflicted by ECMO that initiate thrombogenesis, inflammation, and platelet activation, which exacerbate HR, are of equal significance. To determine the true effect of HR, immune activation can be assessed through the simultaneous measurement of white blood cells and the platelet count, C3b, innate immunity cytokines, and acute phase protein levels. Thrombogenesis, in turn, can be assessed using a coagulogram, viscoelastic hemostatic assays, the functionality of fibrinogen, platelet aggregation, and the products of fibrinolysis. Documenting the flow is also important because several studies report that flow parameters contribute to the extent of thrombogenesis [27]. Raghunathan et al. reports that HR patients had a higher mean flow in comparison to a cohort of the patients without HR [29]. Furthermore, in a study in which ECMO was applied as a bridge to lung transplantation, physicians did not escalate the parameters of VA-ECMO to prevent HR development [69]. In addition, flow rates influence platelet activation and vWF disintegration, which directly influence platelet activity and clearance [4,53,100,103]. Thus, if the documentation of an HR dose of UFH is performed in a similar way going forward, taking into consideration the parameters of inflammation, thrombogenesis, and flow, there is a high probability of identifying a unique dose range for defining HR.

### 7.2. Alternative Anticoagulant

An alternative anticoagulant that has not been highlighted in the Extracorporeal Life Support Organization (ELSO) guidelines [17] or recent reviews [11,18,60] is nafamostat mesilate (NM). This synthetic protease inhibitor is emerging as a noteworthy alternative for patients requiring ECMO support. NM operates by inhibiting serine proteases integral to the coagulation cascade, decreasing thrombin generation and platelet aggregation [127]. Its rapid onset of action, coupled with a relatively short half-life of approximately eight hours, allows for fine-tuned control over anticoagulation, which is particularly beneficial for critically ill patients. The dose typically ranges from 0.46 to 0.67 mg/kg/hour, with maximum reported doses between 1.1 and 2.2 mg/kg/hour in clinical applications [128]. Despite its potential, the use of nafamostat mesilate is tempered by certain limitations. A systematic review has pointed out a scarcity of original studies, resulting in a heterogeneous compilation of data regarding complications, including both bleeding and thrombosis [128]. This calls for further investigation to establish a more comprehensive understanding of its efficacy and safety profile in the context of ECMO.

## 8. Limitations

Inconsistent Definition: The absence of a clear definition for HR complicates search strategies. Differences in study designs and heparin dosing can result in varying outcomes and interpretations.Diversity in Patient Populations: The inclusion of both pediatric and adult patients in the reviewed studies introduces variability in how HR is managed and its incidence, making it challenging to formulate universal conclusions applicable to all ECMO patients.Reliance on the Existing Literature: There is a lack of original studies with proper randomization.There is an inability to simultaneously assess the overall picture of inflammation and coagulation on ECMO due to the dynamic nature of the underlying patient’s disease.

## 9. Summary

HR was shown to be patient- and context-dependent. We illustrated that the dose that defines HR is the maximal UFH dose in the treatment reflecting anti-Xa < 0.3 IU/mL. We also discussed the influence of artificial surfaces on thrombogenesis, immune activation, and platelet consumption, all of which exacerbate HR. We postulate that true HR in patients on ECMO could be determined if all the parameters of inflammation and coagulopathy, taking into account artificial surfaces and the patient’s underlying disease, were to be documented in one randomized clinical study with a pediatric and adult cohort. Considering the dual anti-inflammatory and anticoagulative nature of both heparin and AT, the most appropriate ways to use them should be elucidated. We propose that the definition of HR should include the UFH dose based on the patient’s weight and type of ECMO considering the flow.

## Figures and Tables

**Figure 1 jcm-13-07633-f001:**
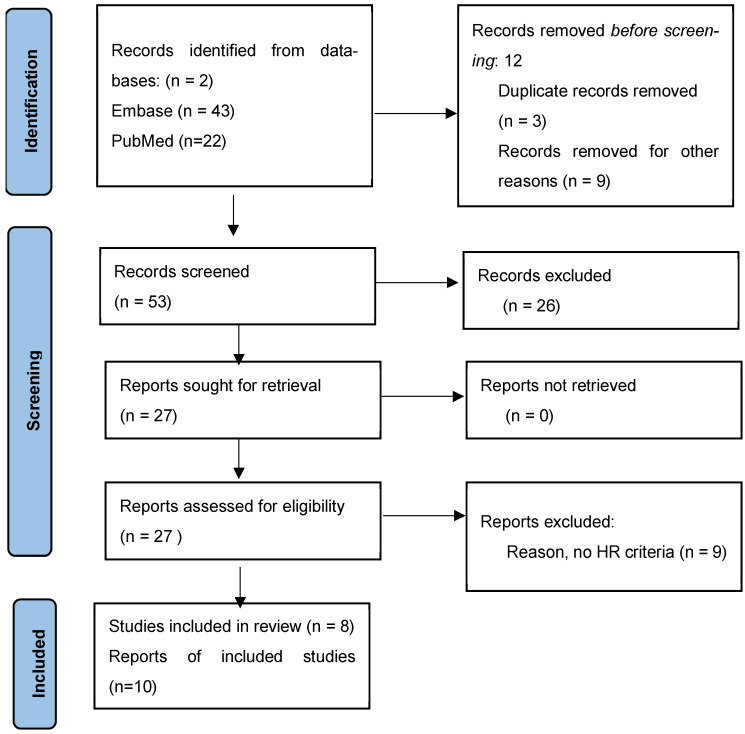
Prisma flow diagram of selection and exclusion strategy.

**Table 1 jcm-13-07633-t001:** Studies describing heparin resistance in patients on extracorporeal membrane oxygenation (ECMO). aPTT—activated partial thromboplastin time; AT—antithrombin; HR—heparin resistance; UFH—unfractionated heparin.

Year and Reference	Study Design and Participants (n)	Type of ECMO and Goal	Traits of Heparin Resistance	HR Incidence (n) and Duration	Other Anticoagulant	Complications
			**Pediatrics**			
Calamaro, 2024 [20]	Case report n = 1 12 y.o.	VA-ECMO Cardiac, respiratory	anti-Xa < 0.35 IU/mL	n = 1 9.5 d	Switch to bivalirudin	Thrombosis on VA-ECMO
Guo, 2024 [21]	Case report n = 1 2 h.o.	VA-ECMO Respiratory	UFH—55 µg/kg/h, anti-Xa < 0.1 U/mL AT = 32% → 50%	n = 1 N/A	Switch to bivalirudin	Hypercoagulation
Procaccini, 2023 [14]	Retrospective n = 191 2 m.o.	ECMO (201 runs): VA—86.6% VV—13.4% Pulmonary 41.3% Cardiac 26.9% ECPR 31.8%	Anti-Xa ≤ 0.3 IU/mL UFH > 40 IU/kg/h UFH ↑ ≥ 2 in 24 h	n = 50 (24.9%) runs 6 d	N/A	Hemorrhage: n = 26 (12.9%) Intracranial hemorrhage: n = 15 (7.5%) Cerebral infarction: n = 18 (9.0%) Circuit clot formation: n = 30 (14.9%) Mortality: n = 112 (55%)
Kaushik, 2023 [22]	Retrospective n = 27 4 m.o.	VA-ECMO 88.9% VV-ECMO 11.1% Pulmonary 18.5% Cardiac 66.7% ECPR 14.8%	Subtherapeutic aPTT UFH = 26 IU/kg/h AT = Normal	n = 1 (3.7%) 6 d	Switch to bivalirudin	Bleeding: n = 12 (44%) Circuit change: n = 3 (11.1%) Mortality: n = 7 (25.9%)
Woods, 2022 [23]	Case series n = 3 12 y.o.	VA-ECMO Cardiac, COVID-19	Subtherapeutic aPTT/Anti-Xa, after AT/FFP	n = 1 (33.3%) N/A	Switch to bivalirudin	Clots in the arterial cannula; circuit change
Hamzah, 2020 [24]	Retrospective n = 16 59 m.o.	VA-ECMO	Subtherapeutic aPTT, Anti-Xa AT < 60%	n = 8 (50%) 114 h	N/A	Thrombosis: n = 3 (18.8%) Bleeding: 5 events/10 d Mortality: n = 5 (31%)
Niimi, 2014 [25]	Case series n = 5 1 m.o.	ECMO	UFH > 40 IU/kg/h, w/no ↑ in ACT	n = 1 (20%) 7 d	N/A	rAT dose is not enough to reach desired AT in pediatric Pts on ECMO
Mejak, 2005 [26]	Case report 3 m.o.	VA-ECMO Cardiac	UFH = 50 IU/kg/h ACT: 120 s	n = 1 Overnight	Switch to argatroban	Thrombocytopenia (HIT), and HIT assay is negative
			**Adults**			
Nagler, 2024 [27]	Retrospective n = 197 Adult	VA-ECMO, VV-ECMO	UFH > 35,000 IU/d (n = 33, 16.8%) UFH > 20 IU/kg/h (n = 14, 7.1%)	n = 47 N/A	N/A	HR is not associated with thrombogenesis (IRR 0.93) Thrombosis is associated with VA-ECMO (IRR, 2.29) and COVID-19 (IRR, 2.33)
Yongpeng, 2022 [28]	Case report n = 1 Adult	ECMO Pulmonary, COVID-19	UFH: > 35 000 IU/d UFH = 43 200 IU/d	n = 1 N/A	UFH + argatroban:	Deep venous thrombosis
Raghunathan, 2021 [29]	Retrospective n = 67 Adult	VA-ECMO 65% VV-ECMO 34%	UFH ≥ 35,000 IU/d anti-Xa < 0.35 IU/mL for VA-ECMO anti-Xa < 0.30 IU/mL for VV-ECMO	n = 34 ≥ 1 d 8.38 ± 5.69 d	N/A	No difference in thrombosis and/or bleeding from non-HR group
Streng, 2020 [30]	Observational n = 3 Adult	VV-ECMO Respiratory, COVID-19	UFH > 35,000 IU/d Anti-Xa > 0.7 IU/mL	n = 3 (100%) 11.3 d	No	Thrombosis; bleeding n = 1 (33.3%)
Gurnani, 2019 [31]	Case report n = 1 Adult (31 y.o.)	VA-ECMO Circulatory failure	ACT < 160 s aPTT 28.5 s Anti-Xa 0.38 IU/mL UFH = 32 IU/kg/h AT = 90%	n = 1 10 d	Switch to bivalirudin	Circuit thrombus
Walker, 2019 [32]	Case series n = 1 Adult (36 y.o.)	VV→VA-ECMO	UFH > 50 IU/kg/h Subtherapeutic Anti-Xa	n = 1 189 h	Switch to bivalirudin	
Sorial, 2019 [33]	Retrospective n = 19 (adult) n = 9 (pediatric)	ECMO Respiratory: 13 (72.2%) Cardiac: 5 (27.8%)	Pediatric: anti-Xa < 0.15 IU/mL UFH = 23.6 IU/kg/h Adult anti-Xa < 0.19 IU/mL UFH = 15.3 IU/kg/h	n = 18 (46.1%) 13.1 d	N/A	Patients with ≥1 major bleed n = 12 (66.7%), >in adult Thrombosis events: n = 45 >in children
Khazi, 2018 [34]	Original n = 42 Adult (50.5 y.o.)	VA-ECMO Cardiac	ACT < 180 ACT < 150 (if bleeding) And/OR UFH > 24,000 IU/d	n = 20 (47.6%) 279.2/176.5 h		Bleeding 80% Survival 41%
Hage, 2019 [35]	Case report n = 1 Adult (41 y.o.)	VV-ECMO Pneumosepsis, ARDS	UFH 61,000 IU < 24 h aPTT = 33 s	n = 1 N/A	Switch to argatroban (2nd day)	Thrombocytopenia (HIT), and HIT assay is negative
Jyoti, 2014 [36]	Case report n = 1 Adult (54 y.o.)	VV-ECMO ARDS	UFH > 30 IU/kg/h + boluses 4000–8000 IU AT = 31% activity	n = 1 23 d	Switch to bivalirudin	Thrombosis, replacement of oxygenator

## Data Availability

Not applicable.

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
