# Peer review of "Heparin Resistance in Patients Receiving Extracorporeal Membrane Oxygenation: A Review"

_jcm, 2024, doi:10.3390/jcm13247633_

Round 1

Reviewer 1 Report

Comments and Suggestions for Authors

Thank you for the detailed review. The topic is well covered in terms of the different coagulation components (origins of HR).

Regarding the population studied, it should be noted that pediatric and adult patients are compared. On the other hand, veno-venous (low flow) and veno-arterial (high flow) ECLS systems are compared. As is also pointed out in the paper, the activity of AT varies with age. However, the underlying diseases treated (e.g. ARDS after aspiration versus central ECMO for vitium in infants) are also very different and a comparison is only possible with reservations. In addition, COVID disease per se is highly thrombogenic.

Overall, I was surprised not to find more original studies on heparin resistance.

In the management of HR, in addition to substitution of AT (either directly or indirectly by cryoprecipitate or FFP), a switch to DTI is recommended. Bivalirudin and argatroban have a high risk profile. Substitution of AT is also associated with a high rate of thromboembolism. What are the data on tolerating HR (and continuing therapy with heparin) as long as it does not escalate?

Author Response

Comments 1: Thank you for the detailed review. The topic is well covered in terms of the different coagulation components (origins of HR).

Regarding the population studied, it should be noted that pediatric and adult patients are compared. On the other hand, veno-venous (low flow) and veno-arterial (high flow) ECLS systems are compared. As is also pointed out in the paper, the activity of AT varies with age. However, the underlying diseases treated (e.g. ARDS after aspiration versus central ECMO for vitium in infants) are also very different and a comparison is only possible with reservations. In addition, COVID disease per se is highly thrombogenic.

Overall, I was surprised not to find more original studies on heparin resistance.

Response 1: We sincerely appreciate your observations about the distinctions between pediatric and adult patients, as well as the differences between veno-venous and veno-arterial ECLS systems. We wholeheartedly support this amendment and have restructured the incidence table to separate pediatric and adult sections for clarity. This changes be found in the chapter 3, p4.

With regard to the types of ECMO—VA and VV—current EUROELSO guidelines and recent studies do not differentiate between these configurations. We agree that this is an important area that warrants further exploration in future research. 

Additionally, while the guidelines do not provide specific information about antithrombin (AT) immaturity in infants, we have identified relevant data in a few pediatric studies and case reports. Due to immaturity of AT several Hospitals prefers to use Bivalitudin instead of UFH.

We fully agree with your statement. In our review we wanted to show the great variety of UFH dosages that defines HR and existing contribution of inflammation and coagulopaty emerging from extracorporeal circuits, regardless of underlying disease.

Comments 2: In the management of HR, in addition to substitution of AT (either directly or indirectly by cryoprecipitate or FFP), a switch to DTI is recommended. Bivalirudin and argatroban have a high risk profile. Substitution of AT is also associated with a high rate of thromboembolism. What are the data on tolerating HR (and continuing therapy with heparin) as long as it does not escalate?

Response 2: We fully agree with  poor control over DTI, and probability of complications, however recent studies show that DTI may be a safe alternative during ECMO management especially in children (60 works from embase). Some studies have indicated that the use of DTI may offer advantages over Unfractionated Heparin (UFH) management in the context of ECMO.

Regarding on tolerating of HR: we have found data about management of HR without switching to alternative method of anticoagulation with mean value 8.38 ± 5.69 days (34 patients doi: 10.1097/MAT.0000000000001334). The maximum dosage of UFH that can be tolerated. 61,000 IU/day.

Reviewer 2 Report

Comments and Suggestions for Authors

This manuscript provides a comprehensive review of heparin resistance (HR) in patients receiving extracorporeal membrane oxygenation (ECMO). The authors have conducted an extensive literature search and synthesis on this important topic. Overall, this is a well-written and informative review that makes a valuable contribution to the field. However, there are some areas that could be improved:

- The structure of the manuscript could be improved for better readability. Some sections, particularly in the "Coagulation and Inflammation on ECMO" part, could be more clearly organized.

- While the authors provide a good summary of the literature, more critical analysis of conflicting findings or methodological limitations of cited studies would strengthen the review.

- The conclusion section could be expanded to more clearly articulate the key takeaways and future research directions.

- Some figures or tables summarizing key points (e.g., definitions of HR across studies, mechanisms of HR) would enhance the manuscript.

Specific comments:

- The introduction could more clearly state the objectives of the review.

- In section 3, consider presenting the data on HR incidence in a table format for easier comparison across studies.

- The discussion on antithrombin and heparin cofactor II (section 4.1) is informative but could be more concise.

- Section 6 should also include the use of Nafamostat mesilate (doi: 10.1111/aor.14276) as a possible alternative to heparin for ECMO patients.

- The manuscript would benefit from a brief discussion of the limitations of current research in this field.

- Proofread for minor grammatical errors and consistency in abbreviation use.

Author Response

Comments 1: This manuscript provides a comprehensive review of heparin resistance (HR) in patients receiving extracorporeal membrane oxygenation (ECMO). The authors have conducted an extensive literature search and synthesis on this important topic. Overall, this is a well-written and informative review that makes a valuable contribution to the field. However, there are some areas that could be improved:

- The structure of the manuscript could be improved for better readability. Some sections, particularly in the "Coagulation and Inflammation on ECMO" part, could be more clearly organized.

Response 1:  Thank you for pointing this out. We agree with this comment. As a result, we have restructured the content to create two distinct chapters: Chapter 5 p. 10 - 12  "Coagulation on ECMO" and Chapter 6 p. 12 - 14  "Inflammation on ECMO". Thank you for helping us to enhance our work. 

Comments 2: The conclusion section could be expanded to more clearly articulate the key takeaways and future research directions.

Response 2: Thank you for your valuable comment. We fully agree with your insight. We have added points in the conclusion section. This change can be found in the chapter 9, p. 16.

Comments 3: Some figures or tables summarizing key points (e.g., definitions of HR across studies, mechanisms of HR) would enhance the manuscript.

Response 3: We creted graphical representation of AT and UFH consumption on ECMO (attached as a figure). In case if this scheme meets yours expectations we will be happy to it in our manuscript. 

Specific comments 1: The introduction could more clearly state the objectives of the review.

Response: Thank you for your valuable feedback regarding the clarity of the objectives in the introduction. We have revised the section to more clearly state the purpose of our review. The updated introduction now explicitly articulates that the objectives are as follows: "In this review we aimed to comprehensively describe the condition of HR on ECMO from molecular and clinical perspectives. Furthermore, we summarized currently available data on incidence of HR among pediatric and adult cohorts on ECMO. Also, we suggest modification and extension of the current description of HR on ECMO based on collected information". This change can be found in the first chapter 1, page 2.

Specific comments 2: In section 3, consider presenting the data on HR incidence in a table format for easier comparison across studies.

Response 2: Thank you for pointing this out. We agree with this comment.  We have modified the table of incidence structure and subdivided it according to age for pediatrics and adults. This change can be found in the Table 1, chapter 3, page 4-5.

Specific comments 3: The discussion on antithrombin and heparin cofactor II (section 4.1) is informative but could be more concise.

Response 3: Thank you for highlighting these important points. We have incorporated your suggestions and revised the  chapter 4.1 to provide a clearer understanding of the topics discussed. This change can be found in the page 9.

Specific comments 4: Section 6 should also include the use of Nafamostat mesilate (doi: 10.1111/aor.14276) as a possible alternative to heparin for ECMO patients.

Response 4: Thank you for your insightful comments. We have carefully considered your suggestions and added to section 7.1 Alternative antcoagulant. This change can be found in the page 16.

Specific comments 5: The manuscript would benefit from a brief discussion of the limitations of current research in this field.

Response 5:  Thank you for your valuable comment, we fully agree. we have added section: Limitations of the review.  This change can be found in the chapter 8. page 16.

Specific comments 6: Proofread for minor grammatical errors and consistency in abbreviation use.

Rsponse 6: Thank you for pointing this out. Our manuscript was edited by MDPI author service. The final version of the manuscript is scheduled for additional editing to ensure its quality.

Reviewer 3 Report

Comments and Suggestions for Authors

Dear author,

This manuscript summarises the literature on heparin resistance (HR) during ECMO support, highlights the discrepancies in the definition and management, discusses the 13 papers found in the literature on this topic, and suggests a specific management approach. Congratulations on the great efforts!

I will start my review by asking you about the scientific added value you provided in this manuscript; on the one hand, there are no specific and uniform guidelines for defining and dealing with anticoagulation and HR during ECMO support, which makes it different for each patient. On the other hand, you suggested a "patient-dependant" treatment approach, which is obvious and acceptable. Still, you divided your management suggestion into younger than 3-6 months and all the other patients! As an ECMO service provider, I can assure you that ECMO patients are all unique, and patient-dependant treatment is the main approach worldwide.

However, I agree with the "patient-dependant" suggestion and I agree with the need for specific guidelines on this topic.  

Second, it would be more reliable to split the cohort into two main cohorts: adults and pediatrics. As you stated, the coagulation cascade (AT maturity) is different in newborns and adults, and ECMO management is quite different.

Third, it would be more reliable to include a flowchart of the literature search you have done.    

Author Response

Comments 1: Dear author,

This manuscript summarises the literature on heparin resistance (HR) during ECMO support, highlights the discrepancies in the definition and management, discusses the 13 papers found in the literature on this topic, and suggests a specific management approach. Congratulations on the great efforts!

I will start my review by asking you about the scientific added value you provided in this manuscript; on the one hand, there are no specific and uniform guidelines for defining and dealing with anticoagulation and HR during ECMO support, which makes it different for each patient. On the other hand, you suggested a "patient-dependant" treatment approach, which is obvious and acceptable. Still, you divided your management suggestion into younger than 3-6 months and all the other patients! As an ECMO service provider, I can assure you that ECMO patients are all unique, and patient-dependant treatment is the main approach worldwide.

However, I agree with the "patient-dependant" suggestion and I agree with the need for specific guidelines on this topic.  

Response 1: Thank you for your valuable feedback regarding the scientific value of our manuscript. We have tried to gather information for deep understanding of the nature of heparin resistance (HR)  on the malecular level. We also tried to collect information about current incidence of HR, which was not met in the studies before. 

Comments 2: Second, it would be more reliable to split the cohort into two main cohorts: adults and pediatrics. As you stated, the coagulation cascade (AT maturity) is different in newborns and adults, and ECMO management is quite different.

Response 2: Thank you for pointing this out. We fully agree with this comment.  We have modified the table of incidence structure and subdivided it according to age for pediatrics and adults.  This change can be found in the Table 1, chapter 3, page 4-5. We also changed the text describing the frequency of HR and its complications. Changes can be found in Chapter 3, page 6-7.

Comments 3: Third, it would be more reliable to include a flowchart of the literature search you have done. 

Response 3:   Thank you for your valuable comment, which will undoubtedly help improve our manuscript. We fully agree with your insight. As a result, we have added a flowchart with the strategy of literature search. This change can be found in the chapter 3, page 3.

Round 2

Reviewer 2 Report

Comments and Suggestions for Authors

The manuscript improved significantly after the revision. I have no more comments to make.

Author Response

Comment: The manuscript improved significantly after the revision. I have no more comments to make.

Response: Thank you very much! We appreciate your contribution into the improvement of our manuscript.

Reviewer 3 Report

Comments and Suggestions for Authors

Dear author,

Thank you for the efforts correcting the manuscript according to the suggestions however:

1) The flowchar is still very poorly organized and misleading. You did not explain the reason for excluding papers. I suggest using the PRISMA flowchart guide to optimise it. This leads to the next problem which is incomprehensible table1. 

2) Despite trying to split the adults and pediatric cohorts, the table is still mixed as well as the message from this paper. 

Author Response

Dear author,

Comments: Thank you for the efforts correcting the manuscript according to the suggestions however:

1) The flowchar is still very poorly organized and misleading. You did not explain the reason for excluding papers. I suggest using the PRISMA flowchart guide to optimise it. This leads to the next problem which is incomprehensible table1. 

Response: 1  We appreciate your feedback regarding the organization of the flowchart and the need for clarity in the reasons for excluding certain papers. In response to your comment, we have revised the flowchart to align with the PRISMA flow diagram, which has helped enhance its clarity and organization. The updated text and flowchart now clearly outlines the inclusion and exclusion criteria, making it easier to follow the process we undertook in our review. This changes be found in the chapter 3, page 3. Thank you for your valuable suggestion. 

Comments 2: Despite trying to split the adults and pediatric cohorts, the table is still mixed as well as the message from this paper. 

Response: 2 Thank you for your insightful feedback regarding the structure of the table and the clarity of our message in the paper. We understand the importance of distinguishing between the adult and pediatric cohorts for a clearer presentation of our findings.
In response to your comments, we have revised the structure of the table. This changes be found in the chapter 3, p. 4 - 6. We believe that this change enhances the overall clarity and alignment with the main messages of the study.
We hope that the updated table meets your vision and provides a more effective way to present the data. Thank you once again for your valuable suggestions, which have helped to improve our manuscript.